# Glycoprotein and Lipoprotein Profiles Assessed by 1H-NMR and Its Relation to Ascending Aortic Dilatation in Bicuspid Aortic Valve Disease

**DOI:** 10.3390/jcm12010332

**Published:** 2022-12-31

**Authors:** Borja Antequera-González, Marta Faiges, Neus Martínez-Micaelo, Laura Galian-Gay, Carmen Ligero, María Ferré-Vallverdú, Lluís Masana, Núria Amigó, Arturo Evangelista, Josep M. Alegret

**Affiliations:** 1Group of Cardiovascular Research, Pere Virgili Health Research Institute (IISPV), Universitat Rovira i Virgili, 43204 Reus, Spain; 2Cardiology Department, Hospital General Universitari Vall d’Hebron, VHIR, CIBER-CV, 08035 Barcelona, Spain; 3Cardiology Department, Hospital Universitari Sant Joan de Reus, Universitat Rovira i Virgili, 43204 Reus, Spain; 4Vascular Medicine and Metabolism Unit, IISPV, Hospital Universitari Sant Joan de Reus, Universitat Rovira i Virgili, 43204 Reus, Spain; 5Biosfer Teslab SL, DEEEA, Metabolomics Platform, Universitat Rovira i Virgili, IISPV, CIBERDEM, 43007 Tarragona, Spain

**Keywords:** bicuspid aortic valve, BAV, lipoprotein metabolism, remnant cholesterol, ascending aortic dilatation, glycoprotein, HMR

## Abstract

Introduction: The bicuspid aortic valve (BAV) confers a high risk of ascending aorta dilatation (AAoD), although its progression seems highly variable. Furthermore, the implication of lipoprotein metabolism and inflammation in the mechanisms that underlie AAoD is not fully established. The aim of this study consisted of evaluating the impact of the lipoprotein and glycoprotein profiles in AAOD as well as its progression in BAV aortopathy. Methods: Using 1H-nuclear magnetic resonance (1H-NMR), we analyzed and compared the lipoprotein and glycoprotein profiles of plasma samples from 152 BAV patients with dilated and nondilated ascending aorta. Additionally, these profiles were also compared for 119 of these patients who were prospectively followed-up clinically and by echocardiography in the long-term (5 years). Ascending aorta dilation velocity (mm/year) was calculated for this analysis. Results: Several parameters related to the lipoprotein profile including remnant cholesterol, small LDL and IDL-cholesterol were found to be significantly increased in the dilated group compared to those in the nondilated group. The glycoprotein A-nuclear magnetic resonance (NMR) signal, a novel inflammation biomarker, was also observed to be increased in the dilated group. After performing multivariate analysis, remnant cholesterol remained an independent variable related to AAoD. In the long-term follow-up, proatherogenic lipoprotein parameters were related to ascending aorta dilatation velocity ascending. After a lineal regression analysis, non-HDL particles remained as an independent predictor of ascending aorta dilation velocity. Conclusions: Patients with BAV and AAoD presented a more pro-atherogenic profile assessed by 1H-NMR, especially related to triglyceride-rich lipoproteins. This pro-atherogenic profile seems to contribute to the higher growth rate of ascending aorta diameter.

## 1. Introduction

Bicuspid aortic valve (BAV) disease is the most common congenital cardiac malformation, with a prevalence between 0.6 and 2% in the general population [1,2,3,4]. In this condition, two cusps are identified in the aortic valve instead of three, usually because two of them are fused by a raphe [5]. This alteration produces a high risk of valvular dysfunction such as stenosis or regurgitation and aortic dilatation [6,7]. In fact, ascending aorta dilatation (AAoD) might affect up to 50–77% of BAV patients [8,9]. A significant AAoD confers a higher risk of aortic dissection or rupture.

The mechanisms that underlie AAoD are not yet definitively clear. The proposed causes include anomalous flow in the ascending aorta generated by the odd dynamics of BAVs and genetic abnormalities responsible for the defective structure of the aortic media, to which cardiovascular risk factors such as hypertension may contribute [1,8,10,11,12,13]. Additionally, some studies have suggested that oxidative stress and consequently, endothelial dysfunction, may be present in BAV disease [14,15,16]. In our previous work, we showed a decreased antioxidant plasma metabolic profile in BAV individuals with AAoD [17].

Furthermore, it is not fully clear whether lipoprotein metabolism could play a key role in AAoD progression. While lipoprotein metabolism seems to be directly related to abdominal aortic dilatation, its role in AAoD, and specifically in BAV disease, remains unknown [18,19]. Although an association between low-density lipoprotein (LDL) and apolipoprotein B (ApoB) and aortic dilatation in BAV patients has previously been highlighted by our group, the research in this area remains insufficient [20]. Hence, an extensive analysis that comprises different lipoprotein subclasses seems necessary to obtain new insights into lipoprotein metabolism and its relation to AAoD in BAV disease.

The aim of this study was to analyze the complete lipoprotein and glycoprotein profiles using ^1^H-nuclear magnetic resonance spectroscopy (^1^H-NMR) in a BAV population with a long-term prospective follow-up, assessing its relationship with the rate of AAoD. This information would provide important physio pathological information that could potentially help to predict a faster rate of AAoD; therefore, possible complications could be prevented such as aortic aneurysm and dissection by improving treatment and medical follow-up.

## 2. Materials and Methods

### 2.1. Study Participants

This study included those patients recruited in our BAV database who were older than 18 years, followed-up for 5 years, and with a plasma sample stored in our biological sample bank at the baseline. Patients with Marfan syndrome were excluded. The participants were prospectively entered into the database and underwent an echocardiogram and blood sample collection upon enrollment. Once patients were included, a clinical and echocardiographic follow-up was performed at least once a year by the same cardiologist in a monographic consultation. Written informed consent was obtained from all patients who participated in this study. BAV disease was diagnosed when two aortic leaflets were clearly visualized, with or without raphe, on the parasternal short-axis view of a transthoracic echocardiogram. If it was inconclusive, transesophageal echocardiogram or cardiac magnetic resonance imaging was performed according to the clinical criteria to obtain a definitive diagnosis. The valve morphotype (i.e., the pattern of cusp fusion) was categorized as typical (fusion between right and left coronary cusps) or atypical (other morphotypes, mainly fusion between right and noncoronary cusps). The maximum diameter at the end-diastole was selected as the ascending aorta measure. All echocardiographic studies were performed or reviewed upon inclusion by the same observer. An exhaustive medical history including cardiovascular risk factors, a complete physical examination, and anthropometry data were recorded.

At the baseline, the patients were divided into two groups: dilated (ascending aorta diameter ≥40 mm) [21] and nondilated (ascending aorta diameter <40 mm) groups. The baseline measurement of the ascending aorta was compared with that recorded in the 5-year follow-up control (range ±1 year). The ascending aorta dilation velocity during follow-up was calculated considering the difference in the ascending aorta diameter between the first and last echocardiograms in the follow-up and dividing it by the follow-up duration (mm/year) for the longitudinal analysis. Severe aortic regurgitation was defined according to an integrated approach considering qualitative, semi-quantitative, and quantitative parameters [22]. Severe aortic stenosis was defined when the mean aortic gradient was ≥40 mmHg or when the aortic valve area was ≤1 cm^2^ [22]. Patients with an ascending aorta diameter ≥50 mm were excluded because they could be considered for surgery. For patients who underwent aortic valve or ascending aorta replacement, the last follow-up before surgery was selected. We included 152 patients who met the established criteria at baseline, who constituted the transversal analysis group. From them, 33 had an ascending aorta diameter ≥50 mm or underwent cardiac surgery before the 5-year follow-up. Therefore, 119 patients constituted the longitudinal analysis group. 

This study was conducted according to the principles of the Declaration of Helsinki and approved by the Institutional Review Board and Ethics Committee (03-06-19/6proj4 and 114/2020) of our institutions, the Hospital Universitari Sant Joan and the Institut d’Investigació Sanitària Pere Virgili.

### 2.2. Plasma Preparation

Blood samples were collected overnight under fasting conditions and processed within 90 min of collection. The samples were centrifuged at 1500× *g* for 15 min to obtain plasma, which was further centrifuged at 4000× *g* for 10 min to remove platelets. The plasma was stored at −80 °C in our biological sample bank (Biobanc IISPV—HUSJR) until needed.

### 2.3. ^1^H-NMR Lipid and Glycoprotein Profile Evaluation

The lipoprotein profile was measured in serum samples (250 μL) using the ^1^H-NMR-based Liposcale^®^ test, a new generation nuclear magnetic resonance test by Biosfer Teslab (Reus, Spain). The lipid concentrations (i.e., triglycerides and cholesterol) of the four main classes of lipoproteins (very low-density lipoprotein (VLDL); intermediate-density lipoprotein (IDL), low-density lipoprotein (LDL), and high-density lipoprotein (HDL)), and the particle numbers of nine subclasses (large, medium, and small particle numbers of each of the following: VLDL, LDL, and HDL) were determined as previously reported [23,24]. Briefly, the particle concentrations and diffusion coefficients were obtained from the measured amplitudes and attenuation of their spectroscopically distinct lipid methyl group NMR signals using the 2D diffusion-ordered ^1^H-NMR spectroscopy pulse. The methyl signal was surface fitted with nine Lorentzian functions associated with each lipoprotein subclass: large, medium, and small subclasses of each of the main lipoprotein classes. The area of each Lorentzian function was related to the lipid concentration of each lipoprotein subclass, and the size was calculated from their diffusion coefficient. The different lipoprotein subclasses corresponded to the following diameter size ranges: large VLDL, 68.5 to 95.9 nm; medium VLDL, 47 to 68.5 nm; small VLDL, 32.5 to 47 nm; large LDL, 24 to 32.5 nm; medium LDL, 20.5 to 24 nm; small LDL, 17.5 to 20.5 nm; large HDL, 10.5 to 13.5 nm; medium HDL, 8.5 to 10.5 nm; and small HDL, 7.5 to 8.5 nm. Non-HDL particles are defined as the sum of all lipoprotein particles minus HDL. RemCholesterol is defined by the total cholesterol minus HDL-cholesterol and LDL-cholesterol.

The glycoprotein profile was determined by analyzing the specific ^1^H-NMR spectral region where these protein–sugar bonds resonate (2.15–1.90 ppm) by deconvoluting the spectra by using three Lorentzian functions, as previously reported [24]. For each function, we determined the total area (proportional to concentration), height, position, and bandwidth. The area of glycoprotein A (GlycA) provided the concentration of acetyl groups of protein-bond N-acetylglucosamine and N-acetylgalactosamine, and the area of glycoprotein B (GlycB) provided the concentration of N-acetylneuraminic acid. The glycoprotein F (GlycF) area arises from the concentration in the acetyl groups of N-acetylglucosamine, N-acetylgalactosamine, and N-acetylneuraminic acid unbound to proteins (free fraction). H/W ratios, which reflect the aggregation state of the sugar-protein bonds, were also reported for GlycA and GlycB [25,26]. Height was calculated as the difference from the baseline to maximum of the corresponding NMR peaks, and the width value corresponded to the peak width at half height.

### 2.4. Statistical Analysis

Statistical studies were performed to compare the distribution of the different variables in a transversal analysis (at baseline: non AAoD vs. AAoD), and in a longitudinal analysis depending on the ascending aorta dilation velocity (AAoD Velocity). Categorical variables are expressed as percentages, and significant differences were identified using the chi-square test or Fisher’s exact test, as appropriate. The quantitative variables, represented as the mean (standard deviation (SD), were analyzed using the Student’s *t*-test. Pearson’s or Spearman correlation, depending on the distribution, was used to identify the linear relationships between the ascending aorta diameter and the parameters included in the glycoprotein and lipoprotein profiles. Furthermore, a logistic regression with a forward stepwise model was constructed to analyze the independent lipoprotein and glycoprotein variables related to AAoD. Parameters with the strongest correlations to the dependent variable but lower correlation between them were included. In the longitudinal analysis, a linear forward regression was performed. In both multivariate analyses, a forward stepwise method was performed, which helped reduce the number of significant parameters that were selected in the final model. *p* values < 0.05 were considered significant. All analyses were performed using SPSS v25 (IBM Corp., Armonk, NY, USA), and graphs were designed using Prism v9 (GraphPad Software, La Jolla, CA, USA).

## 3. Results

### 3.1. Evaluation of the Relation of Lipoprotein and Glycoprotein Profiles to AAoD in BAV Disease: Transversal Study

#### 3.1.1. Baseline Clinical Characteristics

The study included 152 patients with BAV, predominantly men (72.4%), with a mean age of 47.6 (17.2). The mean ascending aorta diameter at the baseline was 39.1 (7.4) mm, and 69 patients (45.4%) had an AAoD. Severe aortic valve dysfunction was present in 47 patients (30.9%): aortic regurgitation (17.1%), aortic stenosis (12.5%), or both (1.3%).

Baseline characteristics revealed that patients with AAoD were significantly older than patients from the nondilated group (41.7 (14.6) vs. 54.8 (17.3) years; *p* < 0.001). Patients with AAoD had a higher body mass index (BMI) and a higher prevalence of type 2 diabetes and hypertension. Regarding the echocardiographic parameters, a higher aortic valve gradient (mean) (*p* = 0.002) was observed in the dilated group. All data are represented in Table 1.

#### 3.1.2. Lipoprotein and Glycoprotein Profiles Assessed by ^1^H-NMR: Nondilated Vs. Dilated 

The relationship between the lipoprotein and glycoprotein profiles and AAoD evaluated by ^1^H-NMR was assessed. Several variables related to cholesterol and triglycerides, most of them triglyceride-rich lipoproteins as well as glycoproteins, showed a linear relationship with the ascending aorta diameter (Appendix A). The most representative correlations between these parameters and the ascending aorta diameter are shown in Figure 1. The complete correlation values can be seen in the Appendix A.

Our results showed significantly higher concentrations in the dilated group in several parameters regarding cholesterol metabolism including VLDL-C, IDL-C, LDL-C, total cholesterol, and remnant cholesterol (Table 2). Furthermore, our results revealed higher concentrations of several parameters related to triglyceride metabolism in the AAoD group, reflected in numerous parameters, most of them triglyceride-rich lipoproteins and total triglycerides. Patients included in the AAoD group also presented a higher number of VLDL and LDL particles and, specifically, higher concentrations of small VLDL and LDL particles.

Finally, diverse ratios regarding the different lipoprotein compositions were studied. In our study, we found that the IDL-TG/IDL-C ratio was significantly higher in the nondilated group than in the dilated group of BAV patients, whereas the HDL-TG/HDL-C ratio was significantly lower.

The glycoprotein profile was also assessed by ^1^H-NMR and then compared between the two groups (Table 3). Our results showed a significantly higher glycoprotein A NMR signal in the AAoD group. Nevertheless, the height–weight ratio did not show significant differences between the groups. Furthermore, no significant differences were observed in the other measured parameters including glycoprotein F and glycoprotein B or the height/weight ratio.

Logistic regression was used to analyze the relationship between the ascending aorta dilation and diverse parameters, which were differentially expressed between both groups including age, hypertension, type 2 diabetes, remnant cholesterol, Glyc-A signal, LDL-P, BMI, and AV gradient (mean) (Table 4). The model obtained was composed of two independent parameters, remnant cholesterol and age, where both represent a risk for AAoD in BAV patients.

### 3.2. Effect of the Lipoprotein and Glycoprotein Profiles in the Progression of Ascending Aorta Diameter in BAV Disease: Longitudinal Analysis

#### 3.2.1. Ascending Aorta Diameter Progression in BAV Patients: A Long-Term Follow-Up

In order to analyze the lipoprotein and glycoprotein profile impact in the ascending aorta diameter progression in BAV disease, 119 patients from the whole group with a mean age of 48.4 (15.6) years who were prospectively followed-up for 5 years (mean 4.7 (0.7)) were included in a longitudinal analysis. The mean and median aortic dilatation rate were 0.54 (0.93) mm/year and 0.20 mm/year, respectively. Baseline ascending aorta measurement was found to be negatively correlated to AAoD velocity (Figure 2). The complete analysis of the baseline clinical and echocardiographical characteristics can be found in Table 5.

#### 3.2.2. Evaluation of the Lipoprotein and Glycoprotein Profiles’ Influence in AAoD Progression in BAV Aortopathy: A Longitudinal Study

The complete lipoprotein and glycoprotein profiles were also evaluated in relation to the ascending aorta dilation velocity (mm/year) (AAoD Velocity). Several parameters were found to be positively correlated to AAoD Velocity in the lipoprotein profile including IDL-C, LDL-C, LDL-TG, non-HDL-P, and LDL-P as well as its three different sizes (Figure 2). The only parameter negatively correlated to AAoD. Velocity was TG-C/IDL ratio. Complete data can be found in Table 6.

When analyzing the glycoprotein profile, the Glyc-B signal as well as H/W Glyc-B were found to be positively correlated to AAoD. Velocity (Figure 2). The rest of the parameters measured did not show any significant correlation to AAoD Velocity and can be observed in Table 7.

Furthermore, a linear regression was performed in order to identify possible predictors of the AAoD Velocity out of several parameter which were positively or negatively correlated: Ascending aorta (at baseline), LDL-TG, H/W Glyc-B, small LDL-P, IDL-TG, IDL-C and Non-HDL particles and the final model is represented in Table 8 In this analysis, AAo and Non-HDL-P remained as independent parameters in order to determine AAoD Velocity.

## 4. Discussion

In this study, we analyzed the lipid and glycoprotein profiles via ^1^H-NMR in a cohort of BAV patients. We observed that patients with AAoD presented a proatherogenic and proinflammatory profile compared to that in nondilated patients, specified by higher levels of triglyceride-rich lipoproteins and glycA, a biomarker directly related to inflammation [24,27]. Remnant cholesterol, which includes VLDL-C and IDL-C, has emerged as an independent variable related to AAoD in BAV disease. Furthermore, this proatherogenic lipoprotein profile also seemed to contribute to a higher dilation rate of the ascending aorta in the long-term follow-up. In this case, non-HDL particles, a parameter which includes LDL-C, VLDL-C, IDL-C, and Lp(a) cholesterol, have also emerged as an independent variable related to a faster AAoD during follow-up. 

Current knowledge about the pathophysiological mechanisms involved in the dilatation of the ascending aorta observed in BAV disease places the effect of the abnormal flow generated by the irregular opening of the valve as the main cause of the dilation [28]. Unlike the anatomically typical tricuspid aortic valve, which gives rise to central flow, the anomalous opening of the BAV promotes eccentric flow, which determines high asymmetrical shear stresses, which could be decisive in dilation genesis [29]. In addition to these factors, aging contributes to the aortic wall degeneration, and as a consequence, to aortic dilation. Furthermore, there could also be genetic abnormalities in the structure of the aortic wall, which contribute to its predisposition to dilation [30]. In addition to these factors, the added effect that may determine a proatherosclerotic lipoprotein and proinflammatory glycoprotein profile remains unknown.

While wide evidence from epidemiological and experimental studies supports the involvement of dyslipidemia and inflammation in the physiopathology of abdominal aorta dilation, its role in AAoD, specifically in BAV aortopathy, is still uncertain. Our group had previously described an association between LDL/apolipoprotein B and AAoD in BAV disease in a smaller cohort [20]. Nevertheless, in the present study, we deepened the analysis of the relationship between the lipoprotein and glycoprotein profiles and BAV aortopathy-associated AAoD as well as its progression using ^1^H-NMR, a rigorous technique that provides the complete number and size of the different lipoproteins. Our results showed that BAV patients with AAoD had higher levels of lipoproteins with a high content of triglycerides (cholesterol particles rich in triglycerides), conferring a pro-atherogenic and pro-inflammatory profile that would contribute to the progression of AAoD [31]. Different observational and Mendelian randomization studies have described the atherogenicity of TG- and TG-rich lipoproteins [32]. Proposed mechanisms associated with major adverse cardiovascular events and remnant cholesterol include local inflammation as well as atherosclerotic plaque formation [33]. Most of the TG rich particles including small VLDL and IDL and LDL particles easily penetrate into the arterial wall and are then taken up by macrophages, leading to foam cell formation [34]. These mechanisms could be involved in arterial wall damage that would contribute to AAoD. Furthermore, the PESA study revealed that serum TG levels were associated with vascular inflammation as well as subclinical atherosclerosis, irrespective of LDL-C levels [35]. Regarding the abdominal aorta, in addition to LDL, a significant association has been described between abdominal aorta aneurysm and triglycerides, suggesting that this proinflammatory effect of TG could also be involved in the development of aortic dilation [36].

The finding of the relation of non-HDL particles with a faster ascending aorta dilation velocity in the longitudinal study is consistent with that observed in the transversal analysis. Non-HDL particles have been highlighted by many to be an excellent cardiovascular risk biomarker, instead of LDL-C [37,38,39]. The number of particles, rather the amount of the cholesterol, has repeatedly been demonstrated to be a better CVD marker when these two parameters are discordant [40]. In addition to LDL, non-HDL includes the TG-rich lipoproteins that constitute the remnant cholesterol. Therefore, our results in the transversal and in the longitudinal study suggest the potential role of the pro-atherogenic lipoproteins contributing to the ascending aorta dilation in BAV patients, highlighting the effect of the TG-rich lipoproteins. This led us to hypothesize that the co-occurrence of hemodynamic alterations and lipidomic factors in BAV patients could simultaneously affect the function of the ascending aorta endothelium, enhancing vascular permeability and therefore accelerating and aggravating the pathological processes that may be caused by each factor individually. 

To understand the possible role of lipoprotein metabolism in the progression of ADD in BAV patients, it is important to consider that although ADD-BAV patients presented higher levels of total cholesterol, LDL-C, and triglycerides than nondilated BAV patients, those values remained within the normal thresholds of the current clinical practice guidelines. This gives rise to the belief that a greater representation of patients with high LDL or TG-rich lipoproteins could have led to a stronger relationship between these parameters and ascending aorta dilation.

The use of echocardiography for measuring aortic diameters instead of computed tomography or cardiac magnetic resonance, techniques with lower variability, is a limitation of our study. However, it must be taken into account that in clinical practice, computed tomography or cardiac magnetic resonance is often not performed in BAV patients without aortic dilatation. Nevertheless, echocardiographic studies have been carried out prospectively with a predetermined methodology and by the same observer, in order to reduce the variability of the measurements. On the other hand, we identified the ascending aorta diameter at the baseline as a variable inversely related to the velocity of ascending aorta diameter progression. Therefore, patients with lower ascending aorta diameters had a higher velocity of ascending aorta diameter progression than patients with higher ascending aorta diameters. The number of patients included in our study was limited and it focused on the glycoprotein and lipoprotein profiles. Therefore, in a larger sample, other variables could be added as variables related to the rate of progression of aortic dilatation. Finally, the study included only BAV patients, so our results cannot be generalized to patients with a tricuspid aortic valve.

Our results suggest the potential benefit of an early implementation of prevention measures aimed at improving the lipidemic profile of BAV patients [20], focused on TG-rich lipoproteins, in order to contribute to slow down the dilation of the ascending aorta. Further studies in the long-term should assess the real impact of these actions.

## 5. Conclusions

We observed that patients with BAV-related AAoD presented a proatherogenic and proinflammatory profile when compared to nondilated patients, specified by higher levels of triglyceride-rich lipoproteins. This proatherogenic lipoprotein profile also seemed to contribute to a higher dilation rate of the ascending aorta in long-term follow-up.

## Figures and Tables

**Figure 1 jcm-12-00332-f001:**
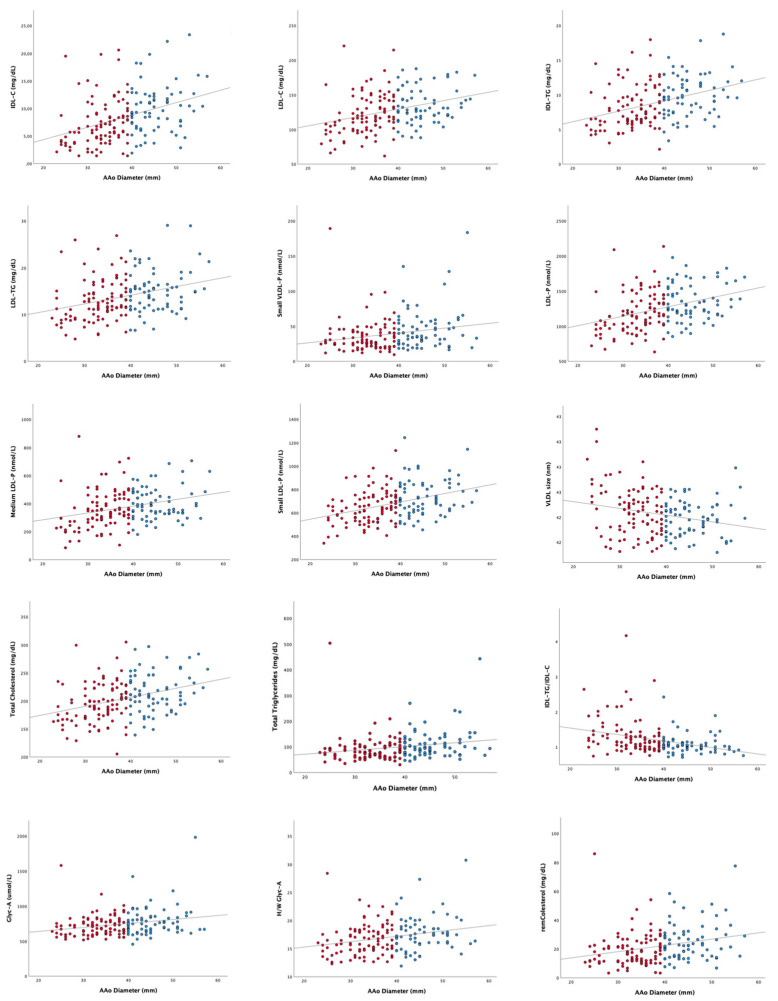
Association between the lipoprotein and glycoprotein profiles and the ascending aorta diameter in the bicuspid aortic valve patients. Red dots correspond to BAV patients without AAD while blue dots correspond to BAV patients with AAD. Complete correlations and *p* values can be found in Appendix A. AAo: ascending aorta; remCholesterol: remnant cholesterol; TG: triglycerides; C: cholesterol; IDL: intermediate density lipoprotein; LDL: low density lipoprotein; VLDL: very low density lipoprotein; *p*: particles; GlycA: glycoprotein A signal; H/W GlycA: height/weight ratio for glycoprotein A signal.

**Figure 2 jcm-12-00332-f002:**
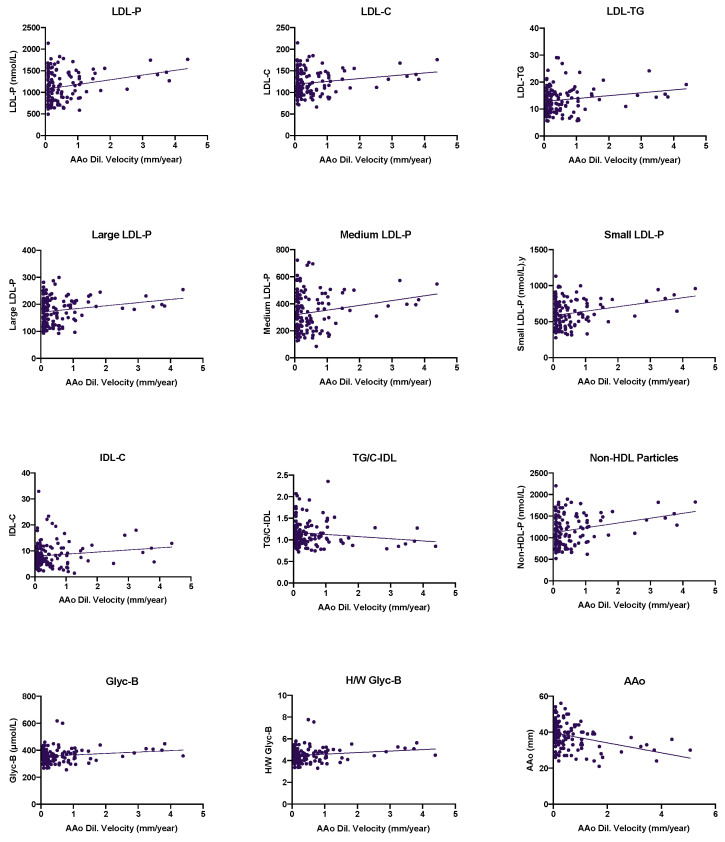
Association between the lipoprotein and glycoprotein profiles and the ascending aorta diameter in the bicuspid aortic valve patients. AAo: ascending aorta (mm) at baseline; AAo Dil. Velocity: ascending aorta dilation velocity; ascending aorta; TG: triglycerides; C: cholesterol; IDL: intermediate density lipoprotein; LDL: low density lipoprotein; P: particles; GlycA: glycoprotein A signal; H/W GlycA: height/weight ratio for glycoprotein A signal.

**Table 1 jcm-12-00332-t001:** Baseline clinical and echocardiographic characteristics of the BAV patients in the transversal analysis, depending on the presence of a dilated ascending aorta.

	NonDIL (n: 83)	DIL (n: 69)	
	Mean (SD) or n (%)	Mean (SD) or n (%)	*p* Value
Sex (male %)	61 (73.5%)	54 (78.3%)	0.430 ***
Age (years)	41.7 (14.6)	54.8 (17.3)	<0.001 ***
Body mass index (kg/m^2^)	24.60 (4.01)	27.53 (4.23)	<0.001 ***
Smoking (%)	34 (41.0%)	26 (37.7%)	0.792
NYHA Scale (≥II)	1 (1.2%)	2 (2.9%)	0.887
Hypertension	19 (22.9%)	36 (52.2%)	0.001 **
Type 2 diabetes	3 (3.6%)	10 (14.5%)	0.024*
Peripheral artery disease	2 (2.4%)	3 (4.3%)	0.517
Stroke	1 (1.2%)	2 (2.9%)	0.465
Coronary artery disease	1 (1.2%)	3 (4.3%)	0.231
Aortic regurgitation (≥2)	34 (41.0%)	30 (43.5%)	0.986
LVEDD (mm)	51.6 (6.1)	52 (5.8)	0.692
LVESD (mm)	31.2 (7.2)	33.2 (6.3)	0.078
Aortic root (mm)	33.9 (5.1)	39.9 (5.7)	<0.001 ***
Ascending aorta (mm)	33.7 (4.1)	45.4 (4.6)	<0.001 ***
AV gradient (mean) (mmHg)	13.1 (13.5)	22 (19)	0.001 **
Statins	9 (10.8%)	10 (14.5%)	0.501
BAV morphology (typical)	47 (56.6%)	45 (65.2%)	0.243

BAV: bicuspid aortic valve; NonDIL: nondilated ascending aorta; DIL: dilated ascending aorta; NYHA: New York Heart Association; LVEDD: left ventricular end-diastolic diameter; LVESD: left ventricular end-systolic diameter; AV Gradient: aortic valve gradient; Typical: right and left coronary leaflet fusion. * Significant values (*p* < 0.05), ** Significant values (*p* < 0.01), *** Significant values (*p* < 0.001).

**Table 2 jcm-12-00332-t002:** Comparison of the lipoprotein profile assessed by HNMR-1 in the transversal analysis, depending on the presence of a dilated ascending aorta.

	NonDIL (n:83)	DIL (n:69)	
	Mean (SD)	Mean (SD)	*p* Value
VLDL-C (mg/dL)	12.08 (8.96)	15.48 (10.39)	0.032 *
IDL-C (mg/dL)	7.76 (4.20)	10.16 (4.35)	0.001 **
LDL-C (mg/dL)	125.17 (30.14)	135.8 (24.69)	0.02 *
HDL-C (mg/dL)	55.08 (10.35)	54.37 (9.06)	0.657
VLDL-TG (mg/dL)	59.48 (50.63)	76.72 (55.01)	0.046 *
IDL-TG (mg/dL)	8.44 (3.02)	10.07 (2.90)	>0.001 ***
LDL-TG (mg/dL)	13.55 (4.48)	15.12 (4.52)	0.033 *
HDL-TG (mg/dL)	9.61 (3.70)	10.84 (3.86)	0.046 *
VLDL-P (nmol/L)	41.33 (30.35)	53.36 (35.37)	0.026 *
Large VLDL-P (nmol/L)	1.07 (0.75)	1.28 (0.74)	0.074
Medium VLDL-P (nmol/L)	5.08 (5.97)	6.36 (5.66)	0.180
Small VLDL-P (nmol/L)	35.18 (24.25)	45.71 (29.45)	0.017 *
LDL-P (nmol/L)	1223.88 (291.67)	1348.36 (253.78)	0.006 **
Large LDL-P (nmol/L)	198.54 (37.91)	208 (33.88)	0.110
Medium LDL-P (nmol/L)	370.51 (139.25)	406.07 (111.42)	0.089
Small LDL-P (nmol/L)	654.83 (142.33)	734.29 (154.03)	0.001 **
HDL-P (µmol/L)	26.26 (5.01)	26.59 (3.87)	0.649
Large HDL-P (µmol/L)	0.26 (0.04)	0.27 (0.04)	0.140
Medium HDL-P (µmol/L)	9.44 (1.74)	9.35 (1.78)	0.754
Small HDL-P (µmol/L)	16.55 (3.85)	16.97 (2.78)	0.455
VLDL-Z (nm)	42.3 (0.24)	42.24 (0.17)	0.080
LDL-Z (nm)	21.13 (0.24)	21.07 (0.25)	0.110
HDL-Z (nm)	8.28 (0.08)	8.26 (0.06)	0.173
TOTAL-C (mg/dL)	200.1 (35.32)	215.81 (33.76)	0.006 **
TOTAL-TGs (mg/dL)	91.08 (55.95)	112.75 (60.14)	0.023 *
VLDL-TG/VLDL-C	5.38 (1.84)	5.22 (1.15)	0.526
IDL-TG/IDL-C	1.23 (0.40)	1.08 (0.27)	0.006 **
LDL-TG/LDL-C	0.11 (0.03)	0.11 (0.02)	0.671
HDL-TG/HDL-C	0.18 (0.07)	0.20 (0.07)	0.024 *
remCholesterol (mg/dL)	19.84 (11.91)	25.64 (13.15)	0.005 **

NonDIL: nondilated ascending aorta; DIL: dilated ascending aorta; remCholesterol: remnant cholesterol; TG: triglycerides; C: cholesterol; HDL: high density lipoprotein; IDL: intermediate density lipoprotein; LDL: low density lipoprotein; VLDL: very low density lipoprotein; P: particles. * Significant values (*p* < 0.05), ** Significant values (*p* < 0.01), *** Significant values (*p* < 0.001).

**Table 3 jcm-12-00332-t003:** Comparison of the glycoprotein profile assessed by ^1^H-NMR in the transversal analysis, depending on the presence of a dilated ascending aorta.

	NonDIL (n:83)	DIL (n:69)	
	Mean (SD)	Mean (SD)	*p* Value
GlycB (µmol/L)	364.99 (52.77)	371.58 (55.55)	0.455 0.123 0.031 * 0.470 0.068
GlycF (µmol/L)	236.35 (78.75)	257.74 (91.11)	
GlycA (µmol/L)	723.67 (154.61)	789.09 (214.47)	
H/W GlycB	4.59 (0.67)	4.67 (0.70)	
H/W GlycA	16.87 (2.77)	17.74 (3.08)	

Non-DIL: nondilated ascending aorta; DIL: dilated ascending aorta; H/W: height/weight. * Significant values (*p* < 0.05).

**Table 4 jcm-12-00332-t004:** Multivariate analysis of the variables related to ascending aorta dilation in the transversal analysis.

		95% CI of OR		
	OR	Inf.	Sup.	*p* Value
Age (year)	1.041	1.017	1.065	0.001 **
Remnant cholesterol (by mg/dL)	1.042	1.007	1.079	0.019 *

* Significant values (*p* < 0.05), ** Significant values (*p* < 0.01).

**Table 5 jcm-12-00332-t005:** Baseline clinical and echocardiographic characteristics of the BAV patients in the longitudinal analysis, depending on the ascending aorta dilation velocity.

	Ascending Aorta Dilation Velocity (mm/Year)	
	Pearson/Spearman Correlation	*p* Value
Sex (male %)	0.044	0.622
Age (years)	0.062	0.491
Body mass index (kg/m^2^)	0.008	0.933
Smoking (%)	0.024	0.804
NYHA scale (≥II)	−0.004	0.974
Hypertension	−0.036	0.699
Type 2 diabetes	−0.155	0.166
Peripheral artery disease	0.103	0.367
Stroke	0.103	0.368
Coronary artery disease	−0.077	0.504
Aortic regurgitation (≥II)	0.071	0.537
LVEDD (mm)	0.207	0.060
LVESD (mm)	0.121	0.290
Aortic root (mm)	−0.146	0.106
Ascending aorta (mm)	−0.369 **	<0.001
AV gradient (mean) (mmHg)	−0.027	0.765
Statins	−0.057	0.538
BAV morphology (typical)	−0.025	0.835

BAV: bicuspid aortic valve; NonDIL: nondilated ascending aorta; DIL: dilated ascending aorta; NYHA: New York Heart Association; LVEDD: left ventricular end-diastolic diameter; LVESD: left ventricular end-systolic diameter; AV Gradient: aortic valve gradient; Typical: right and left coronary leaflet fusion. ** Significant values (*p* < 0.01)

**Table 6 jcm-12-00332-t006:** Comparison of the lipoprotein profile at baseline assessed by ^1^H-NMR in the longitudinal analysis depending on the ascending aorta dilation velocity.

	Ascending Aorta Dilation Velocity (mm/Year)	
	Pearson/Spearman Correlation	*p* Value
TOTAL-C (mg/dL)	0.166	0.071
TOTAL-TG (mg/dL)	0.095	0.302
TG/C-VLDL	0.035	0.705
TG/C-IDL	−0.182 *	0.047
TG/C-LDL	0.107	0.248
TG/C-HDL	0.031	0.740
Small VLDL %	0.156	0.090
Small LDL %	0.022	0.812
Small HDL %	0.142	0.123
VLDL-C (mg/dL)	0.088	0.341
IDL-C (mg/dL)	0.184 *	0.045
LDL-C (mg/dL)	0.199 *	0.030
HDL-C (mg/dL)	−0.131	0.155
VLDL-TG (mg/dL)	0.077	0.403
IDL-TG (mg/dL)	0.165	0.073
LDL-TG (mg/dL)	0.216 *	0.018
HDL-TG (mg/dL)	−0.026	0.776
VLDL-P (nmol/L)	0.092	0.321
Large VLDL-P (nmol/L)	0.072	0.436
Medium VLDL-P (nmol/L)	0.038	0.685
Small VLDL-P (nmol/L)	0.101	0.274
LDL-P (nmol/L)	0.269 **	0.003
Large LDL-P (nmol/L)	0.216 *	0.018
Medium LDL-P (nmol/L)	0.226 *	0.014
Small LDL-P (nmol/L)	0.283 **	0.002
HDL-P (μmol/L)	−0.091	0.327
Large HDL-P (μmol/L)	−0.053	0.565
Medium HDL-P (μmol/L)	−0.161	0.081
Small HDL-P (μmol/L)	−0.027	0.770
VLDL-Z (nm)	−0.176	0.055
LDL-Z (nm)	−0.101	0.272
HDL-Z (nm)	−0.134	0.145
Non-HDL-P (nmol/L) remCholesterol (mg/dL)	0.270 ** 0.080	0.003 0.351

remCholesterol: remnant cholesterol; TG: triglycerides; C: cholesterol; HDL: high density lipoprotein; IDL: intermediate density lipoprotein; LDL: low density lipoprotein; VLDL: very low density lipoprotein; P: particles. * Significant values (*p* < 0.05), ** Significant values (*p* < 0.01).

**Table 7 jcm-12-00332-t007:** Comparison of the glycoprotein profile at baseline assessed by ^1^H-NMR in the longitudinal analysis depending on the ascending aorta dilation velocity.

	Ascending Aorta Dilation Velocity (mm/Year)	
	Pearson/Spearman Correlation	*p* Value
Glyc-B (μmol/L)	0.182 *	0.048
Glyc-F (μmol/L)	0.091	0.326
Glyc-A (μmol/L)	0.135	0.144
H/W Glyc-B	0.189 *	0.039
H/W Glyc-A	0.146	0.113

H/W: Height/Weight. * Significant values (*p* < 0.05).

**Table 8 jcm-12-00332-t008:** Linear regression of the variables related to the velocity of ascending aorta dilation in the longitudinal analysis.

		95% C.I		
	ß	Inf.	Sup.	*p* Value
AAo (by mm)	−0.356	−0.073	−0.027	<0.001 ***
Non-HDL-P (by nmol/L)	0.285	<0.001	0.001	0.001 **

AAo: ascending aorta at baseline (mm); Non-HDL-P: non-HDL particles (nmol/L). ** Significant values (*p* < 0.01), *** Significant values (*p* < 0.001).

## Data Availability

The data presented in this study are available on request from the corresponding author. The data are not publicly available due to ethical reasons.

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
