# Peer review of "Glycoprotein and Lipoprotein Profiles Assessed by 1H-NMR and Its Relation to Ascending Aortic Dilatation in Bicuspid Aortic Valve Disease"

_jcm, 2022, doi:10.3390/jcm12010332_

Round 1
Reviewer 1 Report
Dear authors, thank you for submitting your manuscript about the relationship between glycoprotein and lipoprotein and ascending aortic dilatation in bicuspid aortic valve disease.
Basically, I think the paper is interesting, however I have the following comments.
- I wonder how many patients had their aortic diameters documented by echocardiography? Echo gives us clues, but is not a gold standard for determining aortic diameters because of its inaccuracy, e.g., compared to angio-CT. This must be mentioned in limitation.
- Are patients with hereditary primary dyslipidemia excluded from the study?
- please use a uniform abbreviation for ascending aorta dilatation in the manuscript.
- There is an increase of aortic diameters (0.9 mm per year), which means that the patients had a dilatation of 0.45 cm within 5 years. This is not much. We know many factors that can cause such a small dilatation. The most important is age. An aortic dilatation of 45 mm in 5 years could be related mainly to age, naturally influenced by other parameters such as hypertension, dyslipidemia, etc., which should be mentioned in the discussion.
- In addition, aortic diameters should be reported in both groups of dilated and non-dilated aorta at the beginning of the study and at the end.
- It is discussed that dyslipidemia plays a role in aortic dilatation. I wonder if this already known risk is not main reason for dilatation, regardless of bicuspid or tricuspid form of aortic valve? if, dyslipidemia particularly affects the bicuspid valves this should be discussed.
Author Response
I wonder how many patients had their aortic diameters documented by echocardiography? Echo gives us clues, but is not a gold standard for determining aortic diameters because of its inaccuracy, e.g., compared to angio-CT. This must be mentioned in limitation.
First of all, we would like to say thank you for reviewing our manuscript. As said in the Methods section, we present in this manuscript echocardiographic data. In addition to echocardiography, in the clinical practice we perform CT or CMR to BAV patients with a significant ascending aorta dilation. The patients included in this manuscript had a wide range of aortic diameters, from patients without aortic dilation to patients with a wide range of aortic dilation. We believe that combining echocardiographic with CT or CMR measurement will add a bias. The echocardiographic studies were prospectively performed or reviewed by the same observer, reducing the interobserver variability. Nevertheless, we agree with this reviewer that this is a limitation of our study and we have added a limitation paragraph in order to assess these concerns:
We must assume as a limitation the use of echocardiography for measuring aortic diameters instead of computed tomography or cardiac magnetic resonance, techniques with lower variability. However, it must be taken into account that in clinical practice computed tomography or cardiac magnetic resonance is often not performed in BAV patients without aortic dilatation. Nevertheless, the echocardiographic studies have been carried out prospectively with a predetermined methodology and by the same observer in order to reduce the variability of the measurements.
Are patients with hereditary primary dyslipidemia excluded from the study?
No, only Marfan syndrome patients were excluded. However, we did not identify any patient with very high levels of total cholesterol or LDL-cholesterol (outliers).
Please use a uniform abbreviation for ascending aorta dilatation in the manuscript.
Thank you for your comment. We have revised the manuscript and unify the ascending aorta dilatation abbreviations. AAo: ascending aorta; AAoD: Ascending aorta dilatation; AAoD Velocity: Ascending aorta dilatation velocity.
There is an increase of aortic diameters (0.9 mm per year), which means that the patients had a dilatation of 0.45 cm within 5 years. This is not much. We know many factors that can cause such a small dilatation. The most important is age. An aortic dilatation of 45 mm in 5 years could be related mainly to age, naturally influenced by other parameters such as hypertension, dyslipidemia, etc., which should be mentioned in the discussion.
Thank you for your comment. Indeed, age is an important factor related to aortic dilation in BAV. In the transversal study, age was a variable related to ascending aorta dilation in the univariate and multivariate analysis. Thus, a greater age is related to a higher ascending aorta diameter, and, in consequence, to higher probabilities of aortic dilation. However, in the longitudinal analysis we evaluate the impact of age at baseline on the rate of increase of the ascending aorta diameter. We didn't observe an influence of age. In other words, in this analysis older patients did not dilate quickly than young patients. We have introduced a comment about the effect of aging on ascending aorta diameter on page 12.
In addition to these factors, aging contributes to the aortic wall degeneration and, in consequence, to aortic dilation.
In addition, aortic diameters should be reported in both groups of dilated and non-dilated aorta at the beginning of the study and at the end.
Thank you for this comment. The division of BAV patients into dilated and non-dilated groups has been made in the transversal study. Therefore, we could not present the aortic diameter in the follow-up. In the longitudinal analysis ascending aorta diameter at baseline was inversely related to the rate of progression of ascending aorta diameter. This is presented in table 5 and table 8. Therefore, patients without aortic dilation had a higher velocity of ascending aorta diameter progression than patients with aortic dilation. We have added a specific comment in the Discussion section.
On the other hand, we have identified ascending aorta diameter at baseline as a variable inversely related to the velocity of ascending aorta diameter progression. Therefore, patients with lower ascending aorta diameters had a higher velocity of ascending aorta diameter progression than patients with higher ascending aorta diameters.
It is discussed that dyslipidemia plays a role in aortic dilatation. I wonder if this already known risk is not main reason for dilatation, regardless of bicuspid or tricuspid form of aortic valve? if, dyslipidemia particularly affects the bicuspid valves this should be discussed.
Thank you for this interesting question. We have not analyzed patients with tricuspid aortic valve. So, we have not the answer to this question. We hypothesize that hemodynamic and lipidomic factors in BAV patients could simultaneously affect the function of the ascending aorta endothelium, enhancing vascular permeability and therefore accelerating and aggravating the pathological processes that may be caused by each factor individually. Whether there is a different effect of lipoproteins and pro-inflammatory factors on BAV compared to patients with tricuspid aortic valve should be analyzed in future studies. We added a comment about it in the Discussion section.
Finally, the study included only BAV patients, so our results cannot be moved to patients with a tricuspid aortic valve.
Reviewer 2 Report
This is an interesting study that provides some evidence that individuals with bicuspid aortic valve and ascending aortic dilation exhibited a statistical associated with triglyceride rich lipoproteins and glycoprotein A. The trends appear modest (e.g. Figure 1) and confounded by increased age, levels of type 2 diabetes, and hypertension in the group with dilated ascending aorta, Supplementary Results were missing to enable a full evaluation of the significance of the results. The following points need to be clarified.
1. Indicate the number of potential subjects identified in each group, the number excluded and reasons for exclusion, and any subjects who withdrew from the study.
2. Discuss reasons for specific sample size and sample size power to account for the different hypertension and type 2 diabetes profiles in the two groups.
3. For Figure 1, provide correlation coefficients and p values.
4. For Figure 1, the symbols should be adjusted to distinguish between the individuals with and without dilated ascending aorta.
5. Indicate whether any relation exists between ascending aortic diameter and BMI.
Author Response
This is an interesting study that provides some evidence that individuals with bicuspid aortic valve and ascending aortic dilation exhibited a statistical associated with triglyceride rich lipoproteins and glycoprotein A. The trends appear modest (e.g. Figure 1) and confounded by increased age, levels of type 2 diabetes, and hypertension in the group with dilated ascending aorta, Supplementary Results were missing to enable a full evaluation of the significance of the results. The following points need to be clarified.
Thank you for your kind review. First, we want to apology that supplementary data was not added to the submission due to an error, but it should be added this time.
Indicate the number of potential subjects identified in each group, the number excluded and reasons for exclusion, and any subjects who withdrew from the study.
Thank you for pointing out this. The only exclusion criteria was Marfan syndrome, as described in the Methods section. We included 152 patients recruited in our BAV database who were older than 18 years, with a planned followed-up for 5 years, and with a plasma sample stored in our biological samples bank at baseline. From them, 33 patients were excluded from the follow-up analysis due to an ascending aorta diameter ≥50 mm or because underwent aortic valve or ascending aorta replacement before the planned follow-up (criteria described in the Methods section). Therefore, finally 119 patients were included in the follow-up analysis. We explain better this concept in the Methods section.
Discuss reasons for specific sample size and sample size power to account for the different hypertension and type 2 diabetes profiles in the two groups.
The sample size was enough to identify significant differences in the univariate analysis in the prevalence of hypertension and type 2 diabetes between the two groups, dilated vs non-dilated. Analyzing the theoretical number of patients needed to find the difference obtained, theoretically 57 patients per group would be required in the case of hypertension and 161 patients per group in the case of diabetes, with a beta risk of 0.10 and an alpha risk of 0.05. In any case, no prior estimate of the sample size required to find significant differences in these variables was made. We added a comment in the limitations section regarding the fact that a larger sample size could have influenced whether other significant variables related to dilation or speed of dilation could have been found.
The number of patients included in our study is limited and it is focused on the glycoprotein and lipoprotein profile. Therefore, in a larger sample, other variables could be added as variables related to the rate of progression of aortic dilatation.
For Figure 1, provide correlation coefficients and p values.
These values are now added to figure 1. They are also presented in table S1 in the supplementary data, but it was mistakenly not added to the first submission.
For Figure 1, the symbols should be adjusted to distinguish between the individuals with and without dilated ascending aorta.
Thank you for the suggestion. We have modified this figure in accordance with your suggestion and now BAV patients with non-dilated aorta are represented with red dots, while dilated ones are represented in blue. Complete correlation data is represented in table S1 of the supplementary material.
Indicate whether any relation exists between ascending aortic diameter and BMI.
In the manuscript, we explained that BAV patients with a dilated ascending aorta (>40mm) has statistically significant higher BMI in the univariate analysis. BMI was also positively correlated to ascending aorta diameter with an r:0.326 and p<0.001 (Data not shown). Nevertheless, in the multivariate analysis, BMI did not remain significant in our model.
Reviewer 3 Report
Thank you for inviting me to review this manuscript. This is a reasonably well written manuscript and unique issue. It is a prospective cohort study for investigation of the association between Glycoprotein and lipoprotein profile and ascending aortic dilatation in bicuspid cohorts.
After careful reading of the manuscript, some following considerations can be made:
Comment line 48: “or” in stead of “and”.
Comment lin2 53: Typo.
Author Response
Thank you for inviting me to review this manuscript. This is a reasonably well written manuscript and unique issue. It is a prospective cohort study for investigation of the association between Glycoprotein and lipoprotein profile and ascending aortic dilatation in bicuspid cohorts.
Thank you very much for your kind review.
After careful reading of the manuscript, some following considerations can be made:
Comment line 48: “or” in stead of “and”.
Thank you for pointing out this mistake. It is corrected in the new version of the manuscript.
Comment lin2 53: Typo.
Thank you for your comment but we were unable to find any typos in the line you wrote.